# Evaluating Mulch Cover with Coir Dust and Cover Crop with Palma Cactus as Soil and Water Conservation Techniques for Semiarid Environments: Laboratory Soil Flume Study under Simulated Rainfall

**Abelardo A.A. Montenegro [1], Thayná A.B. Almeida [1], Cleene A. de Lima [1], João R.C.B. Abrantes [2,3] and João L.M.P. de Lima [2,3,*]**

[1] Department of Agricultural Engineering, Federal Rural University of Pernambuco, Rua Dom Manoel de Medeiros, Dois Irmãos, Recife, PE 50910-130, Brazil; abelardo.montenegro@ufrpe.br (A.A.A.M.); thayna.britoalmeida@ufrpe.br (T.A.B.A.); cleene.lima@unitri.edu.br (C.A.d.L.)

[2] MARE—Marine and Environmental Sciences Centre, Department of Life Sciences, Faculty of Sciences and Technology, University of Coimbra, Rua da Matemática, 49, 3004-517 Coimbra, Portugal; jrcbrito@student.uc.pt

[3] Department of Civil Engineering, Faculty of Sciences and Technology, University of Coimbra, Rua Luís Reis Santos, Pólo II-Coimbra University, 3030-788 Coimbra, Portugal

[*] Correspondence: plima@dec.uc.pt

**Abstract:** This paper aims to investigate the performance of mulch cover with coir dust (*Cocos nucifera* L.) and cover crop with Palma cactus (*Opuntia ficus indica* Mill.) as soil and water conservation techniques, in a laboratory soil flume under simulated rainfall. Palma cactus plants oriented at 90° and 30° angles with the slope direction were considered. Simulations comprised uniform advanced and delayed rainfall patterns. Runoff hydrographs and soil loss were monitored at the downstream end of the flume. Soil moisture and flow velocity were measured, and several hydraulic parameters of runoff were estimated. Results show that both mulch cover with coir dust and cover crop with Palma cactus were effective in reducing runoff and soil loss and increasing soil moisture content, thus being both suitable soil and water conservation techniques for semiarid environments. Coir dust was more effective than Palma cactus. Palma cactus oriented at a 90° angle was slightly more effective than Palma cactus oriented at a 30° angle. Differences between advanced and delayed rainfall patterns on the hydrological and erosive response were more pronounced for the mulch cover condition, where no runoff and soil loss were observed at the downstream end of the flume for the advanced rainfall pattern.

**Keywords:** mulching; vegetative barrier; plant orientation; rainfall patterns; runoff and soil loss

## 1. Introduction

Soil and water loss are major global environmental problems in agricultural and rural lands of semiarid regions (e.g., rainfed cropping systems of the northeast region of Brazil), characterized by soils with low infiltration rates and irregular storm patterns, with high-intensity and low-frequency rainfall events occurring mainly in the beginning of the rainy season when soil is more susceptible to evaporation and erosion [1–4]. Soil and nutrients losses are a main threat for agricultural lands, reducing soil fertility, land productivity and eventually leading to the unsustainability of agricultural production systems.

For a long time, soil and water conservation practices have been used to reduce runoff and soil losses, prevent land degradation and to improve the fertility and productivity of agricultural soils. Such subject has been addressed based on detailed hydro-meteorological measurements, monitoring campaigns and modelling activities in well-instrumented experimental rural catchments, such as the ones belonging to the Brazilian Semiarid Hydrological Network, REHISA [5–9]. Such field studies under natural rainfall provide for an excellent opportunity to improve the understanding of the actual impact of distinct agricultural conservation practices on water security and soil protection. However, collecting representative data on runoff and soil loss under natural conditions in semiarid environments is very challenging, as the number of rainfall runoff events is usually very limited. Also, field studies, particularly those under natural rainfall conditions, are typically very time-consuming and demanding in resources, as they often require many years to produce representative results. Therefore, the use of rainfall simulators in laboratory experiments using soil flumes [7,10–14] and field experiments using erosion plots [7,15–17] have been widely used to study runoff and soil erosion processes. Arguably, the main advantage of such experiments, namely using laboratory soil flumes, is that they allow systematic replication of a wide range of rainfall and/or terrain conditions (e.g., rainfall spatial and temporal characteristics, surface slope, soil roughness, initial soil moisture content). Mulch cover [6–8,10,12–14] and cover crop [5,6,8,9] have been shown to provide a higher hydraulic roughness, retarding the surface flow, increasing water infiltration and reducing surface runoff and soil loss, conferring protection to the soil surface from the direct impact of raindrops, thus reducing soil surface crusting, soil compaction, splash erosion and water evaporation and controlling soil temperature fluctuations.

Among the various organic materials used as soil cover, coir-based solutions exhibit one of the highest levels of efficiency in controlling runoff and soil loss [18–20]. Coir is a waste by-product from the coconut industry, obtained from de-husking the coconut fruit (*Cocos nucifera* L.), and its disposal can cause environmental problems in many producing countries, namely in Brazil, which is one of the highest producers in the world. Coir is one of the least expensive natural fibers and has one of the highest tensile strengths. Also, they present high durability and slow biodegradation. In most cases, pre-manufactured commercial geotextile mats of coir fiber are applied over the bare soil surface [18–20]. In contrast, application of coir in the form of loose mulch (e.g., coir dust) evenly distributed over the bare soil surface has not been sufficiently investigated. However, this can be a much cheaper solution.

In recent years, there has been increased interest in Palma cactus (*Opuntia ficus indica* Mill.) for the important role they play in the success of sustainable agricultural systems in marginal areas of arid and semiarid regions, serving both as livestock forage and crop for human consumption [21]. They are well-adapted to arid and semiarid zones characterized by droughty conditions, erratic rainfall and shallow soils subject to erosion, having developed structures to grow and spread under conditions of scarce and erratic rainfall and high temperatures [22]. In the semiarid region of northeast Brazil, cover crop with Palma cactus is being largely adopted as an alternative soil and water conservation technique [5,6,8,9]. Palma cactus presents a modified stem with a thin, flat, racket-shaped vertical structure. Despite its limited rainfall interception, it has a high leaf area index, working as a vegetative barrier to control runoff and soil loss. Despite its considerable application in the field, no laboratory studies have been conducted to evaluate its efficiency as a soil and water conservation technique under simulated rainfall.

The temporal variability of rainfall has a large impact on runoff generation and associated transport processes, particularly in semiarid areas like the northeast semiarid region of Brazil, where storm events span different orders of magnitude [23]. When analyzing the importance of intensity time-fluctuations within a single rainfall event, advanced rainfall patterns with early peaks tend to produce lower runoff and soil losses, and delayed rainfall patterns with late peaks are usually associated with the highest runoff and soil losses. Notwithstanding the large number of experimental studies on this subject using rainfall simulators in the laboratory, e.g., Reference [11], and in the field, e.g., References [15,17], little is still known about the relevance of rainfall patterns under distinct soil cover conditions, and when adopting soil and water conservation methods.

This paper aims to investigate the performance of mulch cover with coir dust and cover crop with Palma cactus as soil and water conservation techniques, in a laboratory soil flume under simulated rainfall. Palma cactus plants oriented at 90° and 30° angles with the slope direction were considered. Their effectiveness in controlling runoff, soil loss, flow velocity and soil water content was evaluated, considering uniform, advanced and delayed rainfall patterns. To our knowledge, this paper presents a first comparative experience in using Palma cactus and coir dust in laboratory soil flume experiments under simulated rainfall.

## 2. Materials and Methods

### 2.1. Laboratory Setup

Laboratory experiments were conducted in the Agricultural Machinery Laboratory of the Agricultural Engineering Department of Federal Rural University of Pernambuco, Brazil, using the rainfall simulator and the rectangular soil flume shown in the photographs of Figure 1.

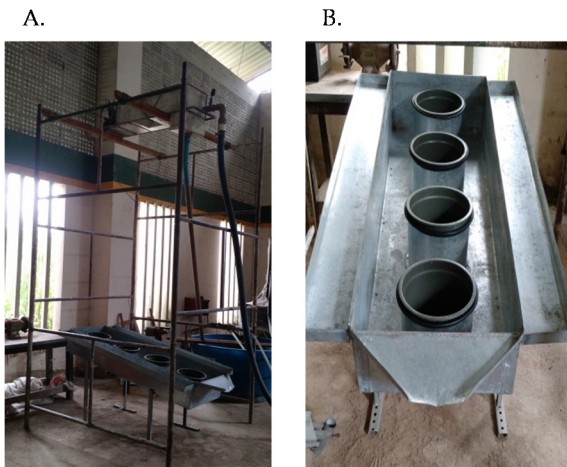

**Figure 1.** Photographs of the laboratory setup used in the experiments: (**A**) Rainfall simulator and corresponding support structure, and (**B**) Downstream view of soil flume with buffer zones.

The rainfall simulator (Figure 1A), used in References [7,9], has an oscillating single downward-oriented flat-spray H–3/8–U–80–100 Veejet nozzle from Spraying Systems Co., with an orifice diameter of 6.2 mm, positioned 2.9 m above the geometric center of the soil flume surface. A 0.75 horse power centrifugal pump supplied the rainfall simulator with tap water at an operating pressure of 0.6 bar at the nozzle, producing a total discharge of 17.5 L min$^{-1}$ with a spray angle of 75°. The uniformity coefficient of rainfall intensity spatial distribution at the soil surface, calculated according to Christiansen [24], was 89%. Raindrops mean diameter, estimated using the flower method [25], was 2.3 mm. The rainfall intensity at the soil surface is adjusted through variation of the oscillating movement of the nozzle, which in turn is controlled by an electrical motor.

The 1.5 m long, 0.5 m wide and 0.2 m deep rectangular soil flume (Figure 1B) was set at a 10% slope gradient, which is representative of experimental field plots located at the representative semiarid watershed from where the adopted soil was collected. Cylindric structures placed at the bottom of the flume were used to provide stability to the cactus plants and distribute them in the flume in different orientations with respect to the slope direction. The flume has a perforated bottom sheet to allow for free drainage of percolated water underneath the soil layer with a thickness of 0.2 m. Two buffer zones (0.15 m width) were fixed at the laterals of the flume to compensate for water and sediments ejected outside the flume due to splash.

A clay loam soil, comprised of 34% sand, 29% clay and 37% silt, was used in the experiments. The original soil was classified as a typical Eutrophic Yellow Argisol, according to Reference [26].

## 2.2. Soil Cover Conditions

Four different soil cover conditions (Figure 2) were considered in this study: (i) One bare soil surface condition (Figure 2A), (ii) one mulch cover condition, with coir dust (*Cocos nucifera* L.) evenly distributed over the bare soil surface, at a density of 8 t ha$^{-1}$ (Figure 2B) and (iii) two cover crop conditions with Palma cactus (*Opuntia ficus indica* Mill.) (Figure 2C,D).

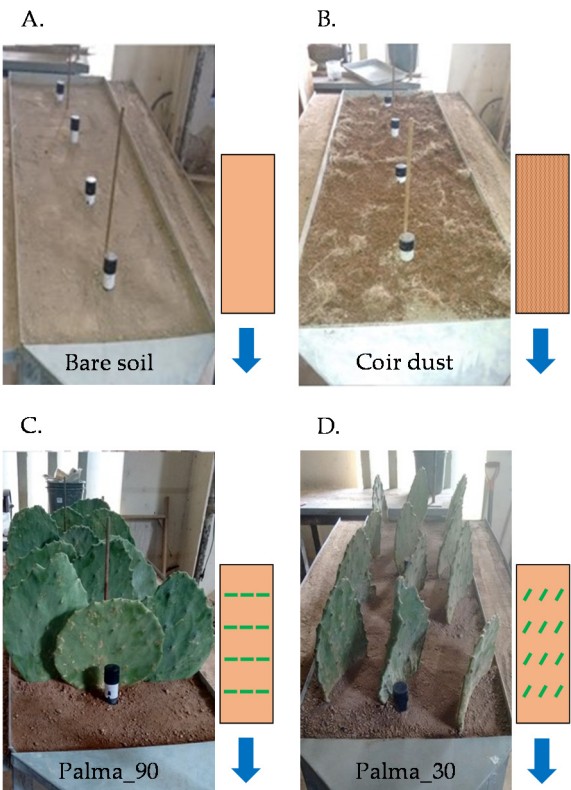

**Figure 2.** Photographs and schematic representations of the four soil cover conditions considered in this study: (**A**) Bare soil, (**B**) mulch cover with coir dust, (**C**) cover crop with Palma cactus plants oriented at a 90° angle with the slope direction (Palma_90) and (**D**) cover crop with Palma cactus plants oriented at a 30° angle with the slope direction (Palma_30). Light brown rectangles represent the soil flume filled with soil. Dark brown dots represent the mulch cover. Green rectangles represent the Palma cactus plants. Blue arrows represent slope direction. Photographs also show the four soil moisture sensors positioned along the flume for each treatment.

For the cover crop experiments, two different arrangements of twelve racket-shaped Palma cactus plants distributed in four contour lines equally spaced of 0.3 m (three plants per line), were considered: (i) Plants oriented at a 90° angle with the slope direction (i.e., transversal to slope direction) (Figure 2C) and (ii) plants oriented at a 30° angle with the slope direction (Figure 2D).

## 2.3. Simulated Rainfall

Three different rainfall patterns were considered in this study (Figure 3): (i) Uniform rainfall pattern with a constant mean intensity of 90 mm h$^{-1}$ and a duration of 15 min, producing a total rainfall amount of 16.9 L, (ii) advanced rainfall pattern, starting with a mean intensity of 90 mm h$^{-1}$ during 5 min followed by a mean intensity of 45 mm h$^{-1}$ during 10 min, producing a total rainfall amount of 11.3 L and (iii) delayed rainfall pattern, starting with a mean intensity of 45 mm h$^{-1}$ during 10 min followed by a mean intensity of 90 mm h$^{-1}$ during 5 min, producing a total rainfall amount of 11.3 L.

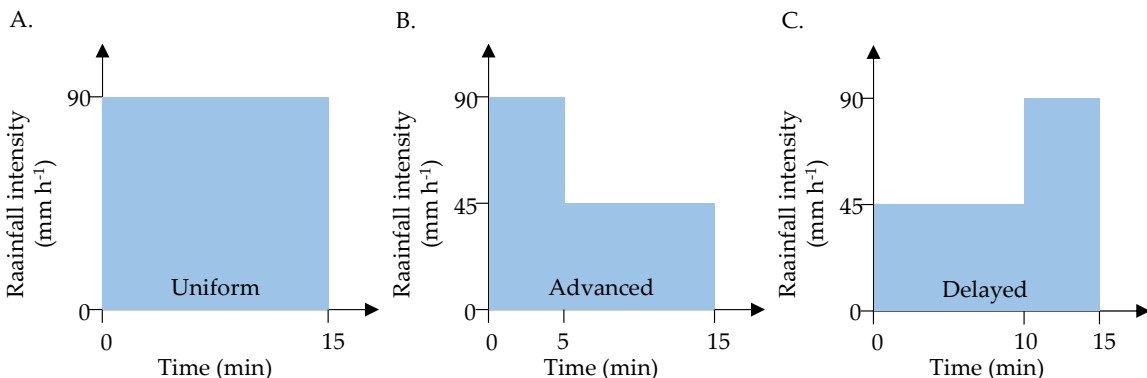

**Figure 3.** Rainfall patterns simulated in the experiments: (**A**) Uniform, (**B**) advanced and (**C**) delayed.

### 2.4. Experimental Procedure

A total of ten scenarios were considered in this study: (i) Nine scenarios combining three different soil cover conditions (bare soil, mulch cover with coir dust and cover crop with Palma cactus plants oriented at a 90° angle with the slope direction) and the three rainfall patterns (uniform, advanced and delayed), and (ii) one scenario with cover crop with Palma cactus plants oriented at a 30° angle with the slope direction, with the uniform rainfall pattern. Experimental runs (i.e., simulation of a rainfall event) for each of the tenscenarios were repeated three times, in a total of 30 experimental runs.

At the beginning of each experimental run, 250 kg of air-dried pre-sieved soil was manually spread over the flume and compacted, to obtain a layer with a uniform thickness of approximately 0.2 m and a bulk density of approximately 1667 kg m$^{-3}$. Then, a sharp straight-edged blade was used to produce a plane and smooth top surface. When mulch cover scenarios with coir dust were tested, 0.6 kg of air-dried pre-sieved coir dust was spread evenly over the soil surface. When cover crop scenarios with Palma cactus were tested, air-dried plants were gently transplanted to the soil in the flume according the desired orientation.

After each experimental run, the remaining soil and coir dust were removed and replaced with new batches, and the Palma cactus plants were gently removed and stored for a next use, following the above-mentioned procedure. This ensured similar initial conditions at the beginning of each experimental run in terms of surface roughness, bulk density and soil moisture content.

### 2.5. Measurements

During each experimental run, runoff hydrographs were monitored at the downstream end of the flume by successive sampling of runoff volumes at regular time intervals of 15 s during the rainfall simulation and 5 s during the runoff recession. Collected samples were dried in a low-temperature oven at 105 °C, allowing soil loss to be evaluated.

Soil moisture was measured in the beginning and in the end of each experimental run, in four points of the soil layer evenly distributed along the longitudinal direction of the flume, using four HidroFarm HFM1010 soil moisture sensors from Falker Ltd. (Figure 2).

Flow velocity at the approximate moment of runoff peak was evaluated using the dye tracer technique, following the procedure described in Reference [27], by measuring the travel time of the leading-edge of a dye tracer from its addition at the upstream end of the flume to its visualization at the downstream end. Flow velocity measurements were only conducted in the experimental runs with simulation of a uniform rainfall pattern.

### 2.6. Hydraulic Parameters of Runoff

The velocity of the leading-edge measured using the dye tracer technique is frequently considered as the surface velocity of the flow and can be regarded as the theoretical maximum (or near maximum)

value of the flow velocity profile. Hence, the actual flow velocity (i.e., theoretical average value of the flow velocity profile) can be calculated from:

$$U = \alpha U_{LE} \tag{1}$$

where U is the flow velocity (m s$^{-1}$), $\alpha$ is the correction factor (–) and ULE is the measured leading-edge velocity of the dye tracer (m s$^{-1}$). The theoretical value of 0.67 for laminar flow reported in Reference [28] was used.

Assuming uniform flow width, flow depth can be calculated from:

$$h = \frac{Q_P}{Ub} \tag{2}$$

where h is the flow depth (m), QP is the runoff peak (m$^3$ s$^{-1}$), U is the flow velocity at the approximate moment of runoff peak (m s$^{-1}$) and b is the flow width (m) assumed equal to the flume width.

Flow regime (i.e., turbulence and speed) was evaluated using the Reynolds and Froude numbers, calculated from:

$$Re = \frac{Uh}{\nu} \tag{3}$$

$$Fr = \frac{U}{\sqrt{gh}} \tag{4}$$

where Re is the Reynolds number (–), Fr is the Froude number (–), $\nu$ is kinematic viscosity of the flow (m$^2$ s$^{-1}$) at a temperature of 20 °C and g is the acceleration of gravity (m s$^{-2}$).

Flow retardation was evaluated using the Manning's roughness and Darcy–Weisbach friction coefficients, calculated from:

$$n = \frac{h^{2/3}S^{1/2}}{U} \tag{5}$$

$$f = \frac{8gSh}{U^2} \tag{6}$$

where n is the Manning coefficient (s m$^{-1/3}$), f is the Darcy–Weisbach coefficient (–) and S is the surface slope (m m$^{-1}$).

### 2.7. Data Analysis

Runoff, soil loss and soil moisture measurements for each experimental run were analyzed for each rainfall profile (uniform, advanced and delayed) as a function of time, comparing the different soil cover conditions under study. As already mentioned, for one of the soil cover conditions (Palma_30), only the uniform rainfall profile was adopted. Hence, a total of ten scenarios were considered in this study. Time to runoff initiation, runoff and soil loss peaks, mean sediment concentrations, total soil loss and runoff coefficients were also calculated. Each scenario was repeated three times, in a total of 30 experimental runs, and mean and standard deviation values were estimated.

### 2.8. Statistical Analysis

The Kruskal–Wallis rank sum non-parametric test for multiple independent samples, followed by a post-hoc multiple comparison Dunn test with *p*-value adjusted according to the family-wide error rate (FWER) procedure of Holm, and then by the false discovery rate (FDR) procedure of Benjamini–Hochberg, were used to examine if: (i) For all rainfall patterns, parameters related to observed runoff and soil loss and observed soil moisture differed significantly among the four soil cover conditions and (ii) for the uniform rainfall pattern, estimated hydraulic parameters of runoff differed significantly among the four soil cover conditions. All statistical analyses were carried out using online freeware ASTATSA [29].

## 3. Results

### 3.1. Runoff and Soil Loss

The temporal evolution of runoff, soil loss and corresponding sediment concentration observed in the experiments are shown in Figure 4. Parameters related to observed runoff and soil loss, together with the results of the statistical analysis, are presented in bar graphs in Figure 5 and summarized in Table 1.

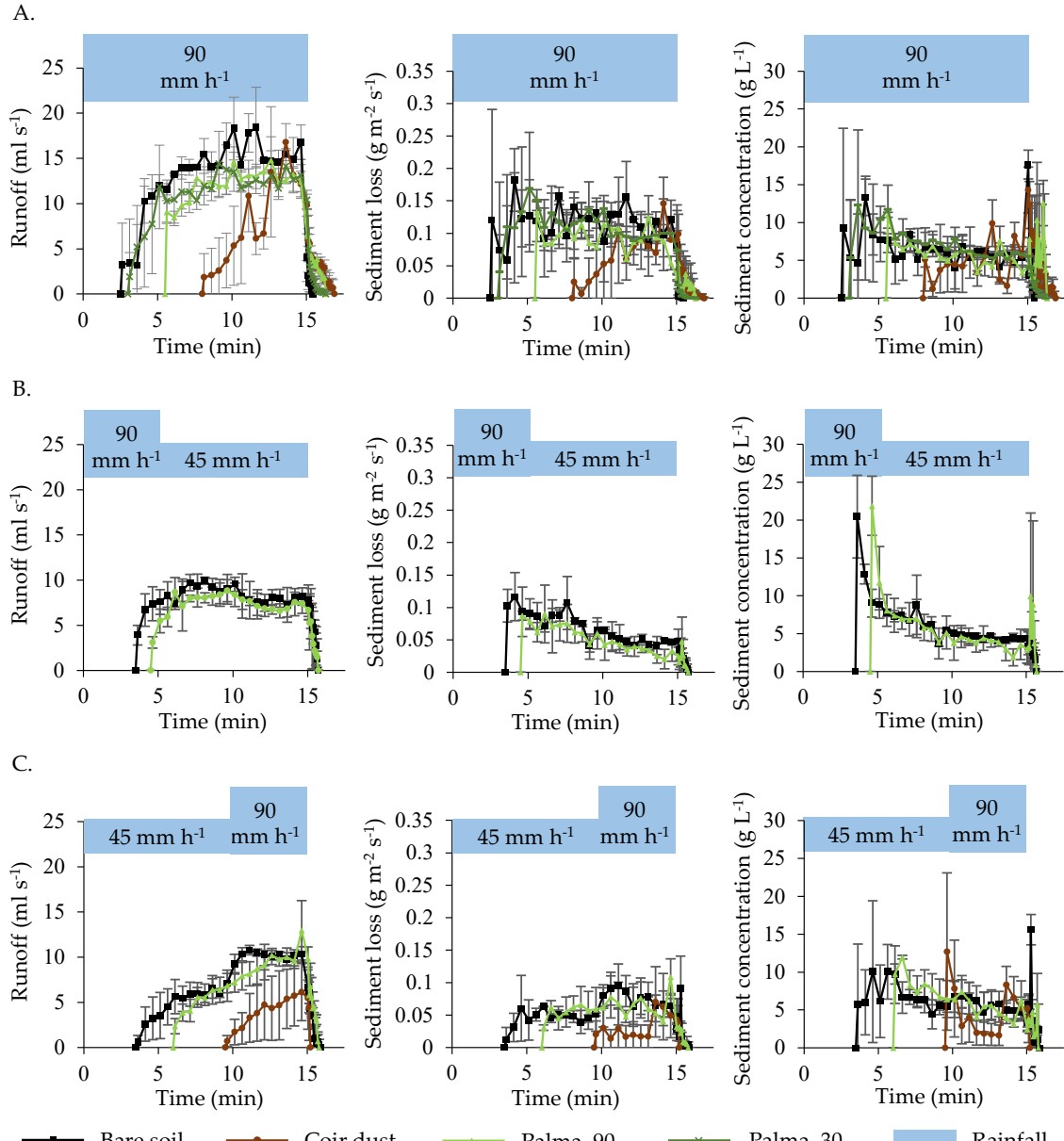

**Figure 4.** Graphs (mean and standard deviation of three repetitions) of runoff (left), soil loss (center) and sediment concentration (right) observed for all soil cover conditions and for the three rainfall patterns simulated in the experiments: (**A**) Uniform, (**B**) advanced and (**C**) delayed. No runoff and soil loss were observed for the coir dust and advanced rainfall pattern.

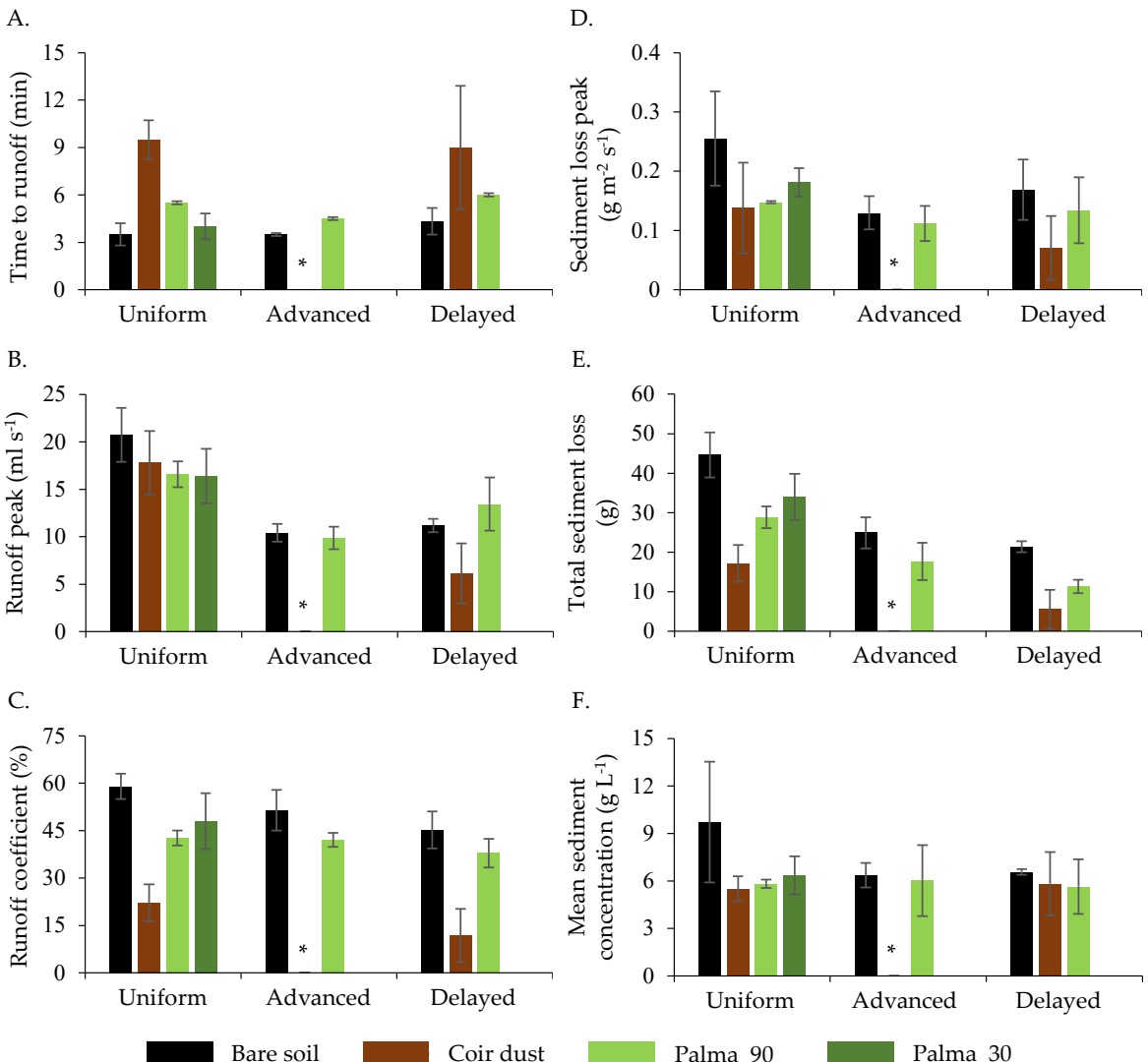

**Figure 5.** Bar graphs (mean and standard deviation of three repetitions) of parameters related to runoff and soil loss observed for all soil cover conditions and rainfall patterns simulated in the experiments: (**A**) Time to runoff, (**B**) runoff peak, (**C**) runoff coefficient, (**D**) soil loss peak, (**E**) total soil loss and (**F**) mean sediment concentration. For the coir dust and advanced rainfall pattern (*), no runoff and soil loss were observed at the downstream end of the flume and time to runoff was not measured.

On average, for the bare soil and all rainfall patterns, runoff at the downstream end of the flume started 3.8 min after rainfall initiation. Cover crop with Palma cactus (Palma_90 and Palma_30) slightly delayed initiation of runoff in 33%, on average. Yet, mulch cover with coir dust more than doubled time to runoff, with a significant increase of 146%, on average, for the delayed and uniform rainfall patterns. For the advanced rainfall pattern, no runoff and soil loss were observed at the downstream end of the flume and, therefore, time to runoff was not measured.

Compared to bare soil, both mulch cover with coir dust and cover crop with Palma cactus reduced runoff and soil loss. Also, both were more effective in reducing soil loss than runoff. However, such reductions were not of the same magnitude. On average, whereas coir dust significantly reduced total runoff in 59%, 100% and 74%, and total soil loss in 61%, 100% and 74% respectively, for the uniform, advanced and delayed rainfall patterns, Palma cactus oriented at a 90° angle (Palma_90) reduced total runoff in only 25%, 18% and 14%, and total soil loss in only 35%, 29% and 47%. Likewise, Palma cactus oriented at a 30° angle (Palma_30) reduced total runoff and total soil loss, for the uniform rainfall pattern, in only 18% and 24%, respectively.

Overall, compared to the aforementioned reductions in total runoff and soil loss, reductions of runoff and soil loss peaks, as well as mean sediment concentration, were less accentuated, except for the coir dust and advanced rainfall pattern, where no runoff and soil loss were observed (i.e., 100% reduction in all parameters related to runoff and soil loss).

**Table 1.** Mean ± standard deviation (of three repetitions) of parameters related to runoff and soil loss observed for all soil cover conditions and rainfall patterns simulated in the experiments. Within the same rainfall pattern, values for a soil cover condition followed by letters in bold ($p < 0.05$) are statistically significantly different from the soil cover condition corresponding to that letter. For the coir dust and advanced rainfall pattern (*), no runoff and soil loss were observed at the downstream end of the flume and time to runoff was not measured.

| Soil Cover Condition | Time To Runoff (min) | Runoff Peak (mL s$^{-1}$) | Runoff Coefficient (%) | Soil Loss Peak (g m$^{-2}$ s$^{-1}$) | Total Soil Loss (g) | Mean Sediment Concentration (g L$^{-1}$) |
|---|---|---|---|---|---|---|
| **Uniform Rainfall Pattern** | | | | | | |
| a. Bare Soil | 3.5 ± 0.7 **b** | 20.7 ± 2.8 | 29.6 ± 2.1 **b** | 0.26 ± 0.08 | 44.6 ± 3.9 **b** | 9.7 ± 3.8 |
| b. Coir Dust | 9.5 ± 1.2 **ad** | 17.8 ± 3.4 | 12.1 ± 2.7 **a** | 0.14 ± 0.08 | 17.2 ± 4.6 **a** | 5.5 ± 0.8 |
| c. Palma_90 | 5.5 ± 0.1 | 16.6 ± 1.4 | 22.2 ± 1.1 | 0.15 ± 0.01 | 28.8 ± 2.7 | 5.8 ± 0.3 |
| d. Palma_30 | 4.0 ± 0.8 **b** | 16.4 ± 2.9 | 24.4 ± 4.5 | 0.18 ± 0.02 | 34.0 ± 5.9 | 6.4 ± 1.2 |
| **Advanced Rainfall Pattern** | | | | | | |
| a. Bare Soil | 3.5 ± 0.1 | 10.4 ± 0.9 | 26.6 ± 3.4 **b** | 0.13 ± 0.03 | 24.9 ± 3.9 **b** | 6.4 ± 0.8 |
| b. Coir Dust | – * | 0 * | 0 * a | 0 * | 0 * a | 0 * |
| c. Palma_90 | 4.5 ± 0.1 | 9.9 ± 1.2 | 21.7 ± 1.5 | 0.11 ± 0.03 | 15.7 ± 4.7 | 6.0 ± 2.2 |
| **Delayed Rainfall Pattern** | | | | | | |
| a. Bare Soil | 4.3 ± 0.9 | 11.2 ± 0.7 | 23.2 ± 2.8 **b** | 0.17 ± 0.05 | 21.4 ± 1.4 | 6.6 ± 0.2 |
| b. Coir Dust | 9.0 ± 3.9 | 6.1 ± 3.2 | 6.1 ± 4.4 **a** | 0.07 ± 0.05 | 5.6 ± 4.9 | 5.8 ± 2.0 |
| c. Palma_90 | 6.0 ± 0.1 | 13.5 ± 2.8 | 19.9 ± 2.3 | 0.13 ± 0.06 | 11.3 ± 1.7 | 5.6 ± 1.7 |

The experiments with uniform rainfall pattern have generated the higher amounts of runoff and soil loss. This is clearly justified by the total depths of simulated rainfall that was 50% higher in the uniform pattern (16.9 L) than in the advanced and delayed rainfall patterns (11.3 L). For the bare soil and Palma cactus oriented at a 90° angle (Palma_90), the advanced rainfall pattern produced slightly lower runoff and soil loss peaks than the delayed rainfall pattern; however, it produced slightly higher total runoff and soil loss. Even so, such differences were never significant. In contrast, and as stated before, for the coir dust, no runoff and soil loss were observed in the advanced rainfall pattern, resulting in a reduction of 100% compared to the delayed rainfall pattern.

## 3.2. Soil Moisture

Volumetric soil moisture before and after rainfall simulation, observed for all soil cover conditions and all rainfall patterns, are presented in Figure 6 and Table 2. Experiments were performed on an initially air-dried soil, which was replaced with a new batch of air-dried soil after each rainfall simulation. On average, this corresponded to a low volumetric soil moisture that only slightly varied between 3.4% and 5.3% for all experiments.

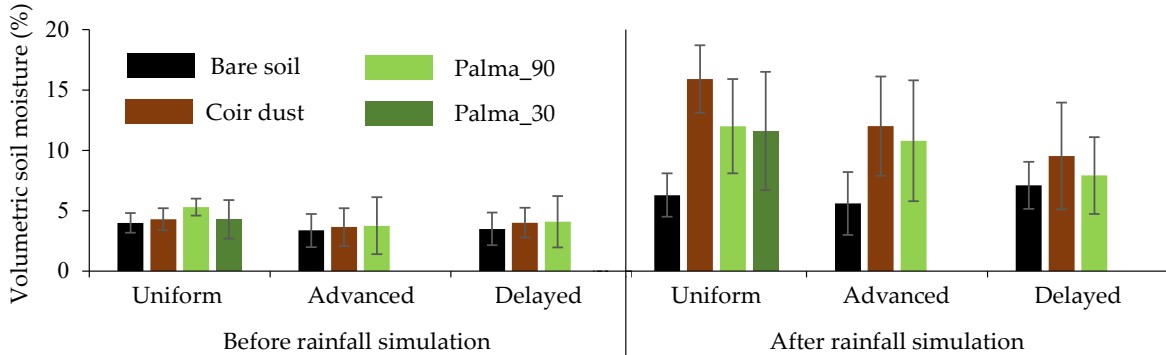

**Figure 6.** Bar graphs (mean and standard deviation of three repetitions) of volumetric soil moisture before and after rainfall simulation observed for all soil cover conditions and rainfall patterns.

**Table 2.** Mean ± standard deviation (of three repetitions) of volumetric soil moisture before and after rainfall simulation observed for all soil cover conditions and rainfall patterns. Within the same rainfall pattern, values for a soil cover condition followed by letters in bold ($p < 0.05$) or bold and underlined ($p < 0.01$) are statistically significantly different from the soil cover condition corresponding to that letter.

| | Volumetric Soil Moisture (%) | |
|---|---|---|
| **Soil Cover Condition** | **Before Rainfall Simulation** | **After Rainfall Simulation** |
| **Uniform Rainfall Pattern** | | |
| a. Bare Soil | 4.0 ± 0.8 | 6.3 ± 1.8 **bcd** |
| b. Coir Dust | 4.3 ± 0.9 | 15.9 ± 2.8 **a** |
| c. Palma_90 | 5.3 ± 0.7 | 12.0 ± 3.9 **a** |
| d. Palma_30 | 4.3 ± 1.6 | 11.6 ± 4.9 **a** |
| **Advanced Rainfall Pattern** | | |
| a. Bare Soil | 3.4 ± 0.9 | 5.6 ± 2.6 **bc** |
| b. Coir Dust | 3.7 ± 1.1 | 12.0 ± 4.1 **a** |
| c. Palma_90 | 3.8 ± 1.3 | 10.8 ± 5.0 **a** |
| **Delayed Rainfall Pattern** | | |
| a. Bare Soil | 3.5 ± 1.3 | 7.1 ± 1.9 |
| b. Coir Dust | 4.0 ± 1.2 | 9.5 ± 4.4 |
| c. Palma_90 | 4.1 ± 1.2 | 7.9 ± 3.2 |

Volumetric soil moisture, observed after rainfall simulation, greatly varied with the soil cover condition. Average volumetric soil moisture values, observed after simulation of the uniform rainfall pattern, were 6.3% for the bare soil (a minor difference compared to 4% observed before rainfall simulation), 12.0% for the cover crop with Palma cactus (Palma_90 and Palma_30) and 15.9% for the mulch cover with coir dust, with the last one being close to the value for the field capacity (16%, according to Reference [9]). The latter two values corresponded to a significantly higher volumetric soil moisture of, respectively, 87% and 152%, when compared to the bare soil. On average, volumetric soil moisture values observed after simulation of the advanced and delayed rainfall patterns were lower. This is mainly attributed to the already mentioned differences in the total depths of simulated rainfall between the uniform pattern and the advanced and delayed patterns. Even so, similarly to the uniform rainfall pattern, on average, volumetric soil moisture values observed after simulation of the advanced rainfall pattern for the Palma_90 and mulch cover were, respectively, 93% and 114% higher than bare soil. For the delayed rainfall pattern, such differences were lower and not significant (11% and 34%, respectively).

### 3.3. Hydraulic Parameters of Runoff

Results of hydraulic parameters of runoff estimated for all soil cover conditions and the uniform rainfall pattern are presented in Figure 7 and Table 3.

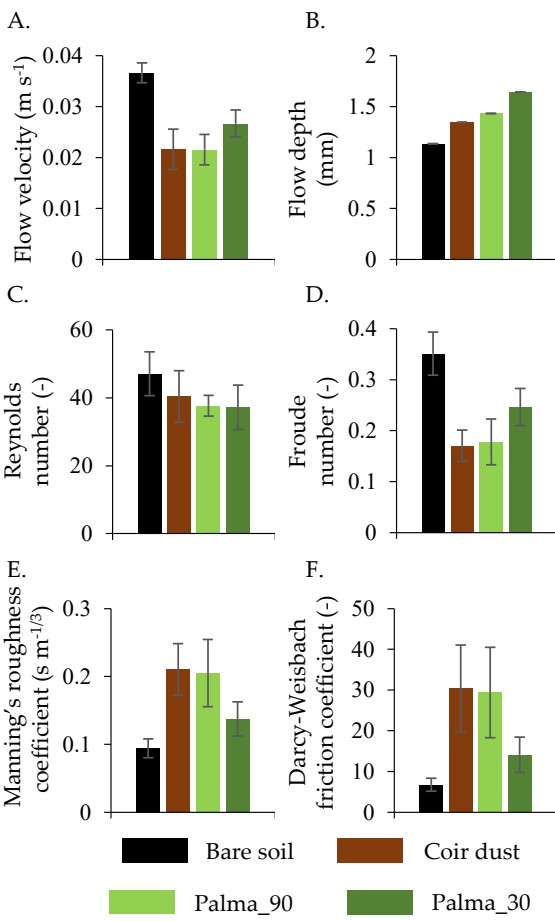

**Figure 7.** Bar graphs (mean and standard deviation of three repetitions) of hydraulic parameters of runoff estimated for all soil cover conditions and the uniform rainfall pattern: (**A**) Flow velocity, (**B**) flow depth, (**C**) Reynolds number, (**D**) Froude number, (**E**) Manning's roughness coefficient and (**F**) Darcy–Weisbach friction coefficient.

**Table 3.** Mean ± standard deviation (of three repetitions) of hydraulic parameters of runoff estimated for all soil cover conditions and the uniform rainfall pattern. Values for a soil cover condition followed by letters in bold ($p < 0.05$) are statistically significantly different from the soil cover condition corresponding to that letter.

| Soil Cover Condition | Flow Velocity (m s⁻¹) | Flow Depth (×10⁻³ m) | Reynolds Number (–) | Froude Number (–) | Manning's Roughness Coefficient (s m⁻¹/³) | Darcy–Weisbach Friction Coefficient (–) |
|---|---|---|---|---|---|---|
| a. Bare Soil | $0.037 \pm 0.002$ | $1.14 \pm 0.19$ | $47.1 \pm 6.5$ | $0.35 \pm 0.04$ **bc** | $0.09 \pm 0.01$ **bc** | $6.8 \pm 1.6$ **bc** |
| b. Coir Dust | $0.022 \pm 0.004$ | $1.35 \pm 0.29$ | $40.4 \pm 7.6$ | $0.17 \pm 0.03$ **a** | $0.21 \pm 0.04$ **a** | $30.4 \pm 10.7$ **a** |
| c. Palma_90 | $0.022 \pm 0.003$ | $1.43 \pm 0.33$ | $37.7 \pm 3.1$ | $0.18 \pm 0.04$ **a** | $0.21 \pm 0.05$ **a** | $29.4 \pm 11.1$ **a** |
| d. Palma_30 | $0.027 \pm 0.003$ | $1.64 \pm 0.03$ | $37.2 \pm 6.5$ | $0.25 \pm 0.04$ | $0.14 \pm 0.03$ | $14.1 \pm 4.3$ |

Compared to the bare soil, both mulch cover with coir dust and cover crop with Palma cactus reduced flow velocity, from an average value of 0.037 m s⁻¹ for the bare soil, to an average value of 0.024 m s⁻¹ for the other soil cover conditions. On average, the lower values of flow depth were

observed for the bare soil and the higher values for the Palma cactus. However, for the flow velocity and flow depth, no significant differences were observed between the different soil cover conditions.

Reynolds and Froude numbers decreased with the coir dust and Palma cactus. For the Reynolds number, differences among soil cover conditions were not significant. On average, Reynolds number ranged between 37.2 and 47.1, which corresponded to a laminar flow (Re < 500), according to the criterion of open channel flow. Froude number varied from 0.35 for the bare soil, to 0.25 and 0.18 for the Palma cactus oriented at 30° and 90° angles (Palma_30 and Palma_90, respectively), to 0.17 for the coir dust. Such values corresponded to a subcritical flow (Fr < 1) according to the criterion of open channel flow. Manning's and Darcy–Weisbach coefficients increased with the coir dust and Palma cactus. For the Froude number, Manning's and Darcy–Weisbach coefficients, only differences between the bare soil and the coir dust and Palma cactus oriented at a 90° angle (Palma_90) were significant.

## 4. Discussion

### 4.1. Coir Dust and Palma Cactus for Soil and Water Conservation

In this laboratory soil flume study under simulated rainfall, mulch cover with coir dust and cover crop with Palma cactus were both effective in reducing runoff and soil loss and increasing soil moisture content. The bare soil exhibited the highest values of runoff and soil loss and the lowest values of soil moisture content. Such results are associated to higher compacting and crusting of the soil surface due to raindrop impact, which reduces infiltration. Coir dust was more effective than Palma cactus. Similar findings were observed in field studies of Lopes et al. [6] and Montenegro et al. [8]. In these field studies [6,8], when compared to the bare soil, reductions of 92% and 74% on mean runoff and soil loss were observed for the mulch cover and significantly smaller reductions of 71% and 54% on mean runoff and soil loss were observed for the Palma cactus. In these field studies [6,8], soil moisture increased by 43% and 23% with the mulch cover and Palma cactus respectively, when compared to the bare soil. Nevertheless, reductions on runoff and soil loss under the two conservation techniques were more pronounced for the field assessment, whereas the increase in soil moisture was higher for the present laboratory soil flume experiments. It is worth to note that Lopes et al. [6] and Montenegro et al. [8] considered a similar Palma application rate and the same mulch cover density of 8 t ha$^{-1}$, although dry grass (*Brachiaria decumbens*) was adopted as mulch cover instead of coir. Also, the analyzed natural rainfall events during the field assessment showed slightly lower mean intensities that the ones adopted in the laboratory study. Despite the low soil thickness of 0.2 m in this laboratory study, lower than the soil profiles in these field studies [6,8], it should be noted that in the laboratory soil flume, the subsurface (infiltrated) water flows downwards towards the perforated bottom, freely draining and, therefore, not affecting runoff generation.

The two techniques for soil and water conservation have a different impact on the hydrological response. Mulch protects soil surface from the direct impact of raindrops, reducing soil compaction and crusting, therefore favoring infiltration, as observed in the field [6–8] and in the laboratory [7,10,12–14]. As observed by Gholami et al. [30], mulch spread on the soil surface can absorb the raindrops' impact on the soil surface, thus preventing soil detachment, maintaining soil surface structure and reducing splash erosion and the transport of soil particle. In contrast, the reduced canopy cover provided by the thin and vertical structure of the Palma cactus gives limited protection to the soil surface and limited rainfall interception [5,6,8,9]. When comparing mulch cover to cover crop for soil erosion control in a semiarid zone of Nigeria, Odunze [31] observed that the better erosion control obtained under straw was attributed to the fact that straw was uniformly spread over the soil surface, and it was close to the soil surface when compared to cover crop whose canopy was further above the soil surface. Mulch cover increases the hydraulic roughness and friction slope of the soil surface, therefore retarding surface flow and enhancing infiltration [7,13]. As observed by Pan and Shangguan [32], increasing cover densities with grass originated a decrease in flow velocity and Froude number and an increase in the Manning's and Darcy–Weisbach coefficients. Moreover, as observed by Reddy [33], the coir dust

can retain and absorb large amounts of water. However, due to its low density, mulch can be easily transported by runoff during high-intensity and/or long-duration rainfall events, exposing unprotected soil surface [6,10,14]. In contrast, Palma cactus provides a more stable natural barrier, retarding surface flow, increasing infiltration [5,9] and inducing sedimentation of soil particles transported by runoff [6,8]. As observed by Bashan et al. [34], desert plants such as Palma cactus are excellent topsoil stabilizers, having the potential to prevent soil erosion and reduce dust in abandoned agricultural areas, if revegetation programs are correctly implemented to allow their growth in degraded areas.

In the short-term, mulch cover with coir dust seems to be a more suitable option than cover crop with Palma cactus to mitigate runoff and soil loss and to promote soil restoration in degraded or vulnerable areas, such as badlands and forest lands following wildfire, and to enhance soil water content and improve agricultural soil fertility and crop productivity in semiarid environments. Generally, compared to cover crop, application of mulch over the soil surface is less time- and money-consuming and has a more immediate effect. Cover crop with Palma cactus can be established either by transplantation or from seeding. Like mulch cover, cover crop by transplantation has a more immediate effect, however it consumes more resources. Cover crop by seeding is much less expensive, however it takes more time to establish and to attain an effect. Also, cover crop establishment by seeding can be disrupted during early growth stages. In the long-term, mulch cover requires more maintenance since it may need to be replaced from time to time due to possible removal by runoff and/or wind and due to expected decomposition. In contrast, in the long-term, cover crop with Palma cactus may present itself as a more permanent runoff and soil loss mitigation technique, that requires lower maintenance and can be more suitable for low-access, non-agricultural marginal areas. Foremost, Palma cactus can serve both as crop for soil and water conservation and as crop for animal and human consumption.

### 4.2. Effect of Palma Cactus Orientation

In this laboratory soil flume study under simulated rainfall, Palma cactus oriented at a 90° and 30° angle with the slope direction exhibited similar hydrological and erosive responses. Even so, due to the favored barrier effect, Palma cactus oriented transversally to the slope direction, close to the contour direction, was slightly more effective in reducing runoff and soil loss. The barrier effect increases flow retardation due to higher hydraulic roughness and friction slope of the soil surface and induces deposition of soil particles on the upslope side of the Palma cactus. It also contributes to a redistribution of runoff, inducing sheet flow with smaller depths, which is less erosive than concentrated flow. In contrast, Palma cactus oriented at a 30° angle presented a low barrier effect, which results in less flow retardation. Also, such orientation results in a lower ability of the Palma cactus to trap soil particles on the upslope side. Furthermore, it may contribute to concentrating flow in the intra-plant spaces, resulting in higher flow depths and a higher erosive effect.

In actual field applications, the orientation of the Palma cactus in relation to the prevailing slope direction can only be controlled if plants are established by transplantation. In this case, a plant orientation of 90° angle with slope direction should be favored to intensify the barrier effect. Yet, it is expected that, most of the time, establishment of cover crop by Palma cactus is done from seeding. Therefore, in this situation, the natural orientation of the plants upon development should be considered when modelling and evaluating the effect of the Palma cactus in the hydrologic and erosive response to rainfall. Ignoring the natural orientation of the plants may lead to unproper estimation of runoff and soil loss.

### 4.3. Comparing Rainfall Patterns

In this laboratory soil flume study under simulated rainfall, for the bare soil and cover crop with Palma cactus, no noticeable differences were observed between advanced and delayed rainfall patterns. However, considering solely the mulch cover with coir dust, meaningful differences were observed between the two rainfall patterns. In contrast to the delayed rainfall pattern, no runoff and

soil loss were observed at the downstream end of the flume for the advanced rainfall pattern. This is in accordance with previous laboratory [11], field [15,17] and numerical [17] analyses, where advanced rainfall patterns, with an early peak, produced lower runoff and soil loss than delayed rainfall patterns with a late peak. Differences between rainfall patterns can be explained by the soil moisture contents and infiltration capacities observed at the time of the different peaks [17]. A delayed peak, occurring later in an event at a moment of higher soil moisture content, results in higher runoff and, consequently, higher soil loss, due to a lower infiltration capacity of the soil. Conversely, an advanced peak, occurring in the beginning of an event at a moment of lower soil moisture content, results in lower runoff and, consequently, lower soil loss, due to higher infiltration capacity of the soil. In the mulch cover with coir dust, this effect was more pronounced due to the capacity of the mulch to retain and absorb water. Like the soil, coir dust at the beginning of an event is more able to retain and absorb water than later in an event.

Despite its limitations, this research suggests that ignoring the intra-variation of rainfall intensity within an event may cause both under- or over-estimation of runoff and soil loss, mainly for soils with high infiltration capacity. This can be even more important when designing soil and water conservation techniques, such as mulch cover, since very often they rely on the capacity of the material (e.g., coir dust) to absorb and retain water in the beginning of an event.

## 5. Conclusions

The main conclusions regarding the laboratory soil flume experiments carried out in this study, under simulated rainfall, to investigate the performance of mulch cover with coir dust and cover crop with Palma cactus as soil and water conservation techniques for semiarid environments were as follows:

- Compared to the bare soil condition, coir dust and Palma cactus were both effective in reducing runoff and soil loss and increasing soil moisture content, making them practical nature-based solutions that can contribute to reclaiming degraded soils and to improving fertility of agricultural soil in semiarid environments.
- Coir dust was more effective than Palma cactus, namely by protecting the soil surface from the direct impact of raindrops, which reduces soil compaction and crusting, therefore favoring infiltration.
- Palma cactus oriented at a 90° angle with slope direction (i.e., transversally to the slope direction) was slightly more efficient than Palma cactus oriented at a 30° angle with slope direction, namely by creating more of a barrier effect, which increases flow retardation and induces deposition of soil particles on the upslope side of the Palma cactus.
- Differences between advanced and delayed rainfall patterns on the hydrological and erosive response were more pronounced on the mulch cover condition, namely by the capacity of the coir dust to retain and absorb water.

Understandably, in quantitative terms, results of this laboratory soil flume study under simulated rainfall should not be extrapolated to actual plot or drainage basin scales and to natural rainfall conditions. Attention should be paid to the specific conditions of this study (e.g., soil characteristics, rainfall intensities, mulch cover density, number of Palma cactus plants) and only a qualitative indication of trends expected for the hydrological and erosive responses should be taken into consideration, and the appropriate upscaling approaches.

**Author Contributions:** Conceptualization, A.A.A.M. and J.L.M.P.d.L.; investigation, T.A.B.A. and C.A.d.L.; formal analysis, T.A.B.A. and J.R.C.B.A.; writing—original draft preparation, C.A.d.L. and J.R.C.B.A.; writing—review and editing, A.A.A.M., T.A.B.A., J.R.C.B.A. and J.L.M.P.d.L.; supervision, A.A.A.M. and J.L.M.P.d.L.; funding acquisition, A.A.A.M. and J.L.M.P.d.L. All authors have read and agreed to the published version of the manuscript.

**Funding:** This research was funded by the Foundation for Science and Technology of Pernambuco State (FACEPE, Brazil) (grant APQ 0300-5.03/17), and by the National Council for Scientific and Technological Development (CNPq, Brazil) (grants 420.488/2018-9 and 446254/2015-0), and was partially funded by the Operational Group

for the Water Management in the Lis Valley Irrigation District—GOLis (project reference: PDR2020-101-030913 (Partner); Partnership nr. 344/Initiative nr. 21).

**Acknowledgments:** The authors acknowledge the technical support from the Agricultural Machinery Laboratory of the Agricultural Engineering Department of Federal Rural University of Pernambuco, Brazil; in particular, from Veronildo Souza de Oliveira.

**Conflicts of Interest:** The authors declare no conflict of interest.

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
