# Peer review of "Evaluating Mulch Cover with Coir Dust and Cover Crop with Palma Cactus as Soil and Water Conservation Techniques for Semiarid Environments: Laboratory Soil Flume Study under Simulated Rainfall"

_hydrology, doi:10.3390/hydrology7030061_

Round 1

Reviewer 1 Report

 This manuscript examines the effect of different mulch cover and different rainfall patterns on runoff and soil loss. The introduction is “straight to the point”, the objectives are well defined and there is a “logic flow” between the methodology, results, discussion and conclusion sections. I recommend this manuscript for publication in hydrology after some minor revisions, given below:

  • Section 2.1: Is there any specific reason for the selection of 10% slope and for the selection of clay loam soil? If yes, please explain. Also, it would be nice to provide the field capacity of the soil and relate it with the soil moisture in the results section.
  • Section 2.5: What was the measuring interval of soil moisture?
  • Discussion: I would like the authors to discuss further on the effect of soil depth on the runoff generation. The soil depth in this experiment is small (20 cm) compared to many "natural" soil profiles, thus I expect that the soil will have less “water storage capacity” which may enhance runoff production. Or not? What about subsurface flow and drainage?

Author Response

Response to Reviewer 1 Comments

Point 1: This manuscript examines the effect of different mulch cover and different rainfall patterns on runoff and soil loss. The introduction is “straight to the point”, the objectives are well defined and there is a “logic flow” between the methodology, results, discussion and conclusion sections. I recommend this manuscript for publication in hydrology after some minor revisions, given below:

Response 1: Thank you for your positive review. Revision was done.

Point 2: Section 2.1: Is there any specific reason for the selection of 10% slope and for the selection of clay loam soil? If yes, please explain. Also, it would be nice to provide the field capacity of the soil and relate it with the soil moisture in the results section.

Response 2: A 10% slope gradient was chosen because it is representative of experimental field plots located at the semiarid watershed from where the adopted soil was collected. At such experimental areas, clay loam soils are dominant.  An additional comment was inserted in the document, addressing this information. As requested, information about field capacity was included at the discussion section. Thank you for the suggestion.

Point 3: Section 2.5: What was the measuring interval of soil moisture?

Response 3: As mentioned in the text, “soil moisture was measured in the beginning and in the end of each experimental run, in four points of the soil layer evenly distributed along the longitudinal direction of the flume, using four HidroFarm HFM1010 soil moisture sensors from Falker Ltd.” Therefore, soil moisture was not measured over the time of the experimental runs. Only point measurements in the beginning and in the end of each experimental run, comprising 2 measurements × 4 points of the soil layer for each experimental run. Firstly, the HidroFarm HFM1010 does not allow for automatic reading at a defined time interval. The operator of the measuring instrument must connect a display and then press a button every time to take a reading from the sensor. Secondly, there was just one measuring instrument for the four sensors buried in the soil.

Point 4: Discussion: I would like the authors to discuss further on the effect of soil depth on the runoff generation. The soil depth in this experiment is small (20 cm) compared to many "natural" soil profiles, thus I expect that the soil will have less “water storage capacity” which may enhance runoff production. Or not? What about subsurface flow and drainage?

Response 4: The flume used in the experiments has a perforated bottom sheet to allow for free drainage of percolated water. Subsurface (infiltrated) water flows downwards towards the perforated flume´s bottom. Therefore, runoff was not affected because we had free drainage of percolated water below the 20 cm. The soil depth in this experiment is small (20 cm) compared to some "natural" soil profiles, but high enough when compared to many soil flume experiments in the literature. Some natural soil profiles can even be shallower. The “water storage capacity” in this setup, per unit depth, was therefore independent of depth.

The following text was added to the Discussion: “The thickness of the soil profile of 0.2 m in this laboratory study can be small when compared to the soil profiles in these field studies [6, 8]. However, it should be noted that in the laboratory soil flume the subsurface (infiltrated) water flows downwards towards the perforated bottom, freely draining and, therefore, not affecting runoff generation.”

Reviewer 2 Report

Comments

SUMMARY

The paper addresses the research area related to water management of the MDPI Hydrology journal. I believe that the target journal is an appropriate forum for this article. The paper aims to investigate the performance of mulch cover with coir dust and cover crop with Palma cactus as soil and water conservation techniques, in a laboratory soil flume under simulated rainfall.

BROAD COMMENT

The Introduction section is well written with recent references. The authors described in detail the methodology used in the study. However, the authors failed to describe in the methodology section how the data were analysed in the study. The authors also failed to discuss the results obtained. In the discussion section, the authors refer to the same studies they used in the introduction section. Besides, another weakness of this study is that the authors failed to repeat the experiment (the second season of runs), therefore, not enough information is provided to establish a strong conclusion to support the findings of the study. I suggest the authors mention this as limitations of the study in the conclusion section.

SPECIFIC COMMENTS

-Line 122: The authors mentioned that the original soil was classified as a typical Eutrophic Yellow Argisol. I suggest the authors mentioned the type of classification (e.g., FAO, ISSS, etc) and also cite the references (sources).

-Lines 204-209: The results presented in Tables 1-3 imply that the authors conducted ANOVA before doing the means separations using post hoc Tukey-Kramer honestly significant difference (HSD). I suggest the authors include in the manuscript the detail about the statistical tests they performed on the data before doing the ANOVA (e.g., normality, homogeneity of variance, etc ). I also recommend them to indicate the software used to do the data analysis.

-Lines 314-393: The authors discussed the results of this manuscript using the same studies (references) that they mentioned in the Introduction section. I suggest the authors discussed in depth the results of their study using additional references to the ones mentioned in the introduction section.

Author Response

Response to Reviewer 2 Comments

Point 1: The paper addresses the research area related to water management of the MDPI Hydrology journal. I believe that the target journal is an appropriate forum for this article. The paper aims to investigate the performance of mulch cover with coir dust and cover crop with Palma cactus as soil and water conservation techniques, in a laboratory soil flume under simulated rainfall.

BROAD COMMENT

The Introduction section is well written with recent references. The authors described in detail the methodology used in the study. However, the authors failed to describe in the methodology section how the data were analysed in the study.

Response 1: Thank you for your positive review. A revision was done. In the Materials and Methods section text was added to explain how the data were analysed.

As mentioned in the text: “2.7. Statistical analysis. One-way analysis of variance (one-way ANOVA) followed by post hoc Tukey-Kramer honestly significant difference (HSD) multiple comparison tests were used to examine if: i) For all rainfall patterns, parameters related to observed runoff and soil loss and observed soil moisture differed significantly among the four soil cover conditions; and ii) For the uniform rainfall pattern, estimated hydraulic parameters of runoff differed significantly among the four soil cover conditions”.

The information on the detail about the statistical tests performed on the data before doing the ANOVA and the software used to do the data analysis were added to the text: “All statistical analyses were done using online freeware ASTATSA [28]. No statistical tests were performed on the data before doing the ANOVA”.

One-way ANOVA (ANalysis Of VAriance) with post-hoc Tukey HSD (Honestly Significant Difference) Test Calculator for comparing multiple treatments. Available online: https://astatsa.com/OneWay_Anova_with_TukeyHSD/ (accessed on 5 August 2020).

Point 2: The authors also failed to discuss the results obtained. In the discussion section, the authors refer to the same studies they used in the introduction section.

Response 2: Discussion section was improved using additional references.

Point 3: Besides, another weakness of this study is that the authors failed to repeat the experiment (the second season of runs), therefore, not enough information is provided to establish a strong conclusion to support the findings of the study. I suggest the authors mention this as limitations of the study in the conclusion section.

Response 3: A total of ten scenarios were considered in this study. Each scenario was repeated three times, in a total of 30 experimental runs. The limitations of the study were included in the conclusion section: Although 30 experimental runs were conducted, more experiments can be conducted with more scenarios including, e.g. more slopes, other types of soils or other mulch types.

Point 4: -Line 122: The authors mentioned that the original soil was classified as a typical Eutrophic Yellow Argisol. I suggest the authors mentioned the type of classification (e.g., FAO, ISSS, etc) and also cite the references (sources).

Response 4: The information on the type of classification was added to the text.

Soil was classified according the Brazilian standard defined in EMBRAPA - Empresa Brasileira de Pesquisa Agropecuária. Manual de Métodos de Análise de Solo; Centro Nacional de Pesquisa de Solos: Rio de Janeiro, Brazil, 1997; 212 pp. (In Portuguese)”. The type of classification and reference were added to the text.

Point 5: -Lines 204-209: The results presented in Tables 1-3 imply that the authors conducted ANOVA before doing the means separations using post hoc Tukey-Kramer honestly significant difference (HSD). I suggest the authors include in the manuscript the detail about the statistical tests they performed on the data before doing the ANOVA (e.g., normality, homogeneity of variance, etc ). I also recommend them to indicate the software used to do the data analysis.

Response 5: In fact, no statistical tests were performed on the data before doing the ANOVA. The one-way ANOVA followed by post hoc Tukey HSD was performed using an online tool available at: https://astatsa.com/OneWay_Anova_with_TukeyHSD/. This online tool has been used widely by the scientific community to perform this statistical analysis, namely by the Medical and Bio-chemical scientific community.

The information on the detail about the statistical tests performed on the data before doing the ANOVA and the software used to do the data analysis were added to the text: “All statistical analyses were done using online freeware ASTATSA [28]. No statistical tests were performed on the data before doing the ANOVA”.

One-way ANOVA (ANalysis Of VAriance) with post-hoc Tukey HSD (Honestly Significant Difference) Test Calculator for comparing multiple treatments. Available online: https://astatsa.com/OneWay_Anova_with_TukeyHSD/ (accessed on 5 August 2020).

Point 6: -Lines 314-393: The authors discussed the results of this manuscript using the same studies (references) that they mentioned in the Introduction section. I suggest the authors discussed in depth the results of their study using additional references to the ones mentioned in the introduction section.

Response 6: Discussion section was improved using additional references.

Round 2

Reviewer 2 Report

I have undertaken a review of the manuscript (revised) as well as the attached author responses to the initial review where I recommended major revisions. I am satisfied with the revisions made by the authors as they have addressed most, if not all, of my initial comments except the following comments:

“The results presented in Tables 1-3 imply that the authors conducted ANOVA before doing the means separations using post hoc Tukey-Kramer honestly significant difference (HSD). I suggest the authors include in the manuscript the detail about the statistical tests they performed on the data before doing the ANOVA (e.g., normality, homogeneity of variance, etc ). I also recommend them to indicate the software used to do the data analysis.”

Since ANOVA is a parametric statistical test, it is imperative to check the normality and the homogeneity of variance. If these conditions are met, one can do the ANOVA, otherwise, you should opt for a non-parametric ANOVA test. Failing to do so, it is likely the results are biased.

Therefore, I do believe that the manuscript will warrant publication in Hydrology if the above conditions are met.

Author Response

Comments and Suggestions for Authors

Point 1: I have undertaken a review of the manuscript (revised) as well as the attached author responses to the initial review where I recommended major revisions. I am satisfied with the revisions made by the authors as they have addressed most, if not all, of my initial comments except the following comments:

“The results presented in Tables 1-3 imply that the authors conducted ANOVA before doing the means separations using post hoc Tukey-Kramer honestly significant difference (HSD). I suggest the authors include in the manuscript the detail about the statistical tests they performed on the data before doing the ANOVA (e.g., normality, homogeneity of variance, etc ). I also recommend them to indicate the software used to do the data analysis.”

Since ANOVA is a parametric statistical test, it is imperative to check the normality and the homogeneity of variance. If these conditions are met, one can do the ANOVA, otherwise, you should opt for a non-parametric ANOVA test. Failing to do so, it is likely the results are biased.

Therefore, I do believe that the manuscript will warrant publication in Hydrology if the above conditions are met.

Response 1:

Dear reviewer. You are correct in your comment. We thank you for your positive review and therefore avoiding an mistake to be printed. Revision was done, where we changed the statistical test used for data analysis, and adapted the text.

The Kruskal-Wallis rank sum non-parametric test for multiple independent samples, followed by post-hoc multiple comparison Dunn test with p-value adjusted according to the family-wide error rate (FWER) procedure of Holm and then by the false discovery rate (FDR) procedure of Benjamini-Hochberg was used to examine if: i) For all rainfall patterns, parameters related to observed runoff and soil loss and observed soil moisture differed significantly among the four soil cover conditions; and ii) For the uniform rainfall pattern, estimated hydraulic parameters of runoff differed significantly among the four soil cover conditions. All statistical analyses were carried out using online freeware ASTATSA [29].

29. Kruskal-Wallis rank sum (omnibus) test calculator with follow-up post-hoc multiple comparison calculator by the methods of (1) Conover, (2) Dunn and (3) Nemenyi. Available online: https://astatsa.com/KruskalWallisTest/ (accessed on 13 August 2020).

Round 3

Reviewer 2 Report

I have undertaken a review of the manuscript (revised) as well as the attached author responses to the initial review where I recommended minor revisions. I am satisfied with the revisions made by the authors as they have addressed most, if not all, of my initial comments. Therefore, I do believe that the manuscript will warrant publication in Hydrology.